REGISTERED REPORT PROTOCOL

# Accuracy of rating scales for evaluating aphasic patients' psychological aspects and language function: A scoping review protocol

Yuhei Kodani[1], Shinsuke Nagami [1]*, Satomi Kojima[2], Shinya Fukunaga[1], Hikaru Nakamura[3]

1 Department of Speech Language Pathology and Audiology, Faculty of Rehabilitation, Kawasaki University of Medical Welfare, Kurashiki, Okayama, Japan, 2 Comprehensive Unit for Health Economic Evidence Review and Decision Support (CHEERS), Research Organization of Science and Technology, Ritsumeikan University, Simogyo-ku, Kyoto, Japan, 3 Department of Contemporary Welfare Science, Faculty of Health and Welfare, Okayama Prefectural University, Souja, Okayama, Japan

* shinsuke.nagami.0514@gmail.com

## Abstract

Aphasia nearly affects half of all poststroke patients. Furthermore, aphasia affects all language functions, well-being, and quality of life of patients. Therefore, rehabilitation of patients with aphasia requires an accurate assessment of language function and psychological aspects. However, assessment scales for language function and psychological aspects of patients with aphasia are said to be inaccurate. In Japan, this sign is more prominent than in English-speaking countries. Therefore, we are putting together a scoping review of research articles published in English and Japanese to date, with the aim of summarizing the accuracy of rating scales for language function and psychological aspects of people with aphasia. The scoping review was intended to be a comprehensive examination of the accuracy of rating scales for people with aphasia. We will search the article databases PubMed, MEDLINE, Embase, PsycINFO, Web of Science, and the Medical Journal Web (Japan). The observational studies that describe the reliability and validity of the rating scales in adult aphasic after stroke will be searched for. There will be no publication date for the articles that will be searched. We believe that this scoping review aims to assess the accuracy of rating scales used to measure different aspects of aphasia, with a focus on research conducted in English-speaking countries and Japan. By conducting this review, we believe to identify any problems with rating scales used in English and Japanese research and improve their accuracy.

## Introduction

Aphasia affects up to 42% of stroke patients [1] and is thought to impair speech, auditory comprehension, reading, and writing abilities [2]. As a result, people with poststroke aphasia (PWA) have lower language function, communication ability, well-being, and quality of life (QOL) [3]. Therefore, rehabilitation, including psychological support, is regarded as important for PWA [4, 5]. One reason for this is that appropriate rehabilitation intervention for PWA is

**Data Availability Statement:** All relevant data from this study will be made available upon study completion.

**Funding:** The authors hereby thank the KAKENHI Grant-in-Aid for Research Activity Start-up (21K2115) for the financial support, duly recognized in the "Funding Statement" section added in the manuscript. Of note, the funders did not partake in the conception of the study, the acquisition and analysis of the data, the decision to publish, or the preparation of the manuscript.

**Competing interests:** The authors have declared that no competing interests exist.

thought to lead to good outcomes and more individualized consequences for them [6, 7]. Rehabilitation providers, such as speech-language pathologists, must accurately assess PWA's language function, communication ability, well-being, and QOL to provide appropriate rehabilitation [8].

PWA language function, communication ability, well-being, and QOL have all been assessed using neuropsychological and psychological rating scales [9]. Many PWA assessment scales are said to be inadequately validated in terms of sensitivity, specificity, measurement reliability, and validity of aphasia-related matter detection, which are indicators of their accuracy [9, 10]. Moreover, little attention has been paid to the accuracy of the PWA assessment scales in Japan. Therefore, assessment scales for the treatment of PWA are currently chosen for subjective reasons, such as ease of use by speech-language pathologists and clinicians, rather than objective reasons, such as accuracy. As an example, in "The Research Outcome Measurement in Aphasia," which attempted to create a standard core outcome set for PWA, the rating scale selection method lacked indices for judging their accuracy. Furthermore, the burden on PWA and rehabilitation providers was considered, and the experts made a subjective decision [11]. The point is that selecting an objective assessment scale for PWA is currently difficult [12]. As a result, there is a risk that speech-language pathologists and clinicians will miss important information that can be used to develop an appropriate rehabilitation plan for PWA and accurately measure the effectiveness of that rehabilitation.

To address this issue, we will conduct a scoping review of the accuracy of rating scales used to assess language function, communication ability, well-being, and QOL in people with PWA in English-speaking countries and Japan. We create a document that compiles all of the recent data on the accuracy of rating scales for PWA. This review is significant because it identifies issues with the accuracy of rating scales for PWA based on all available data. Therefore, it can be useful in planning effective PWA rehabilitation and subsequent research. Although reviews refer to the assessment items in the screening tests for language function with stroke [10], the comprehensive assessment tests for language function, and the assessment tests for aphasia [8, 12], there has been no review of accuracy. Furthermore, no such review studies exist in Japan.

## Objectives

The first purpose of this scoping review is to identify the domains of language function, communication ability, well-being, and QOL of PWA with low accuracy in rating scales based on data from the Japanese database (Ichushi web) and several international databases. The second purpose is to estimate the potential for equalizing the content of rehabilitation for PWA based on the findings of this study. The current study establishes a strategy to equalize the content of PWA rehabilitation. Furthermore, the third purpose is to present differences in assessment scales for PWA in other countries and in Japan and to estimate the rehabilitation intervention for PWA in Japan and the accuracy of the assessment scales. The domain categories in language function, communicative competence, well-being, and QOL were determined during a discussion among the coauthors about the current state of aphasia rehabilitation in Japan and previous studies [11].

## Methods

We chose the scoping review as our study design because it provides a comprehensive view of the PWA rating tests. Additionally, the authors used an online tool (https://whatreviewisrightforyou.knowledgetranslation.net/) to determine which research question was best suited to this study method. Furthermore, our scoping review will be conducted in accordance with the Joanna Briggs Institute's methodology [13]. This consists of six steps: (a)

identifying the research question, (b) identifying relevant studies, (c) selecting previous studies, (d) illustrating the results, (e) analyzing and summarizing the results, and (f) consultation (discussion) with researchers. This study is scheduled to begin in February 2023 and concluded in May 2023.

## Stage 1: Identifying the research question

The purpose of this study was to describe the accuracy of the English and Japanese PWA rating tests. Therefore, we developed the following research question in accordance with this objective:

1. Which of the domains of the aphasia rating scale for PWA (language function, communication ability, well-being, and QOL) has the lowest accuracy?

2. In which domain is the rating scale for PWA in Japanese inaccurate?

## Stage 2: Identifying relevant studies

A comprehensive search of the literature will be conducted using the following electronic databases: PubMed, MEDLINE, Embase, PsycINFO, Web of Science, and Ichushi Web (Japan). As a first step, we conducted a search on PubMed using a search formula developed in consultation with a medical librarian to determine the appropriateness of the search formula and identified 1,788 studies (2022/12/27). A preliminary version of the PubMed search can be found in S1 File, which was developed with the help of a medical librarian experienced in the field (SK).

We plan to search for other databases using similar appropriate terms for each database. In addition, we plan to conduct two searches across all databases. The extraction will be limited to studies in English and Japanese. There will be no publication date for the articles that will be searched. Rayyan, a literature management software, will be used to import all extracted study references, and duplicate studies will be removed.

## Stage 3: Study selection

There will be two screening steps in the study selection process. The first step is to screen literature titles and abstracts, followed by a full-text review. Two independent reviewers (YK and SN) will screen articles against the eligibility criteria in both steps (Table 1). All disagreements will be discussed, and if there is no agreement, a third reviewer (SF and HN) will be consulted.

**Table 1. Inclusion and exclusion criteria.**

| Inclusion criteria | Exclusion criteria |
|---|---|
| The literature is primarily concerned with rating scales aimed at not <20 individuals aged 18 years or older diagnosed with stroke-induced aphasia. This literature comprises observational and empirical studies, as well as research specifically aimed at developing, testing the reliability of and validating rating scales that assess various aspects of poststroke aphasic individuals' language functionality, communication abilities, overall well-being, and QOL. | Literature focusing on aphasia from factors other than stroke (head trauma, Alzheimer's disease, cerebrovascular dementia, dementia with Lewy bodies, primary progressive aphasia, frontotemporal dementia, and progressive supranuclear palsy), literature also focusing on higher brain dysfunction other than aphasia, literature focusing on aphasic under 18 years old, literature focusing on aphasia syndromes of stroke, literature focusing on training for people with aphasia after stroke, literature published before 1995, literature in languages other than English and Japanese, intervention studies, scoping reviews, systemic reviews, meta-analysis, case studies, and letter papers. |

We screened the literature for title and abstract using Rayyan. The first inclusion criterion is the exclusive incorporation of rating scales duly validated with a minimum sample size of 20 participants with PWA aged 18 years or above and supplementation by a cohort with aphasia. The second inclusion criterion is to focus on one or more of the following aspects: language function, communication ability, well-being, and QOL. However, literature focusing on aphasia caused by diseases other than stroke or PWA in children under the age of 18 over will be excluded. Additionally, only observational and empirical studies will be included; intervention studies, scoping reviews, systemic reviews, meta-analysis, case reports, and letter papers will be excluded. Gray literature and conference abstracts will be excluded. Next, we will extract the full text of the eligible studies and double-check the citation details. Moreover, references that do not meet the inclusion criteria will be labeled as "characteristics of excluded references" in the results flowchart, along with the reason for exclusion. For the duration of the process, we will adhere to the PRISMA guidelines [14] (Fig 1).

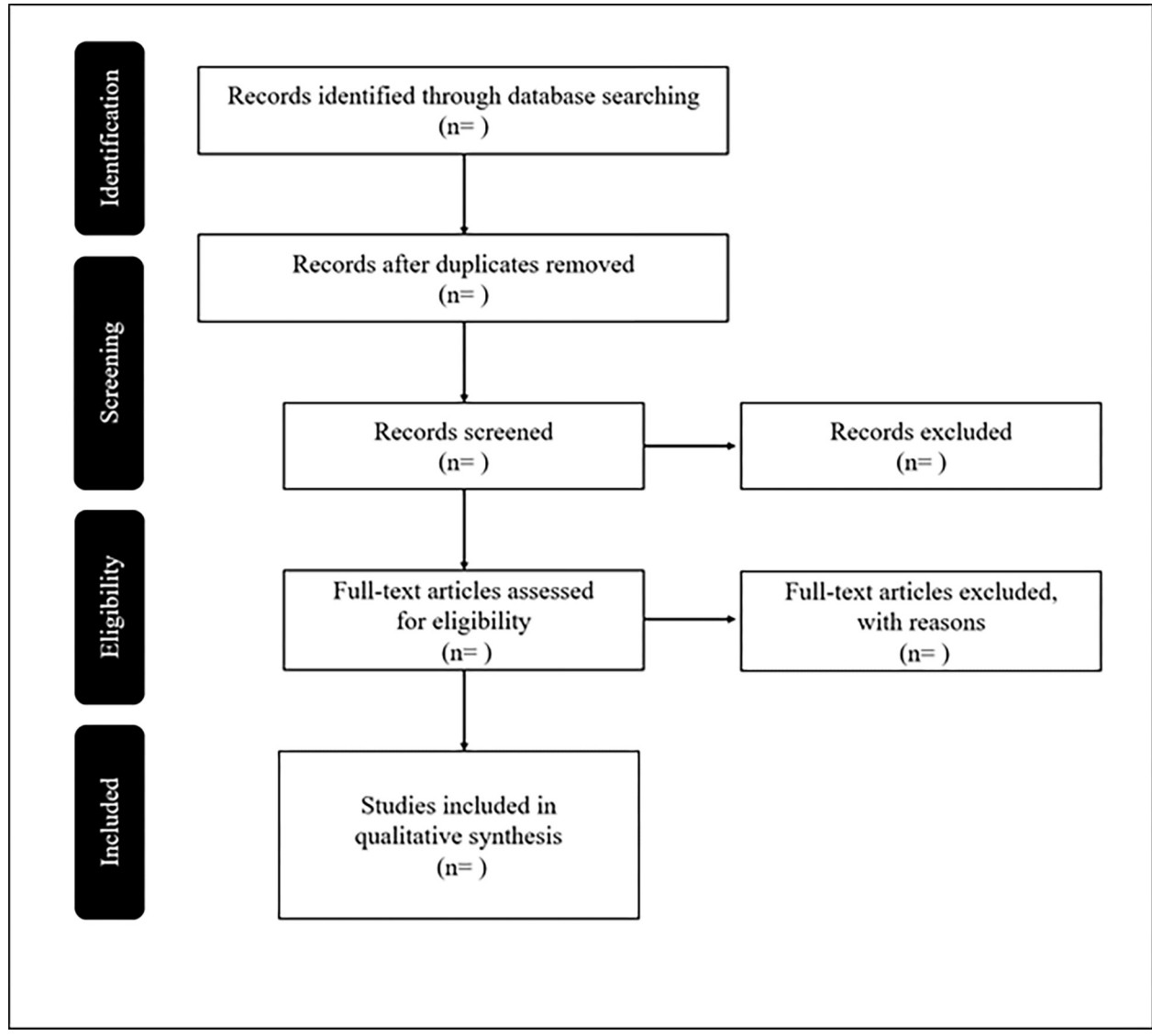

**Fig 1. Flow diagram of study selection process, as depicted by the preferred reporting items for systematic reviews and meta-analysis guidelines.**

### Stage 4: Illustrating the results

Two independent reviewers will extract the relevant results from all included studies in the scoping review. The structured data recording developed by two reviewers(YN and SN) will be entered into Microsoft Excel using the record sheet created by the reviewers. The obtained results encompass a wide array of information, including the author's name, publication year, study title, country of origin, study purpose, sampling method, sample characteristics (sex, age, educational background, presence/absence of prior stroke, and cognitive assessment results), stroke characteristics (stroke type, hemisphere affected, and extent and duration of injury), sample size, PWA assessor's profession, rating scale utilized for PWA evaluation, rating scale domains, details of the rating scale development process, information on rating scale sensitivity and specificity, and information related to the reviews of the rating scale's reliability and validity.

Disagreements between the two reviewers will be resolved through discussion, and any issues that cannot be resolved through discussion will be decided by a third reviewer. Contacting the author(s) will resolve any missing or incomplete research data in the studies.

### Stage 5: Analyzing and summarizing the results

To analyze and summarize the findings of this scoping review, we present a table of the characteristics of the extracted studies. The presented table encompasses various essential aspects, such as the publication year, author's name, pertinent PWA characteristics (e.g., age, gender, education level, length of time since stroke occurrence, stroke type, stroke frequency, hemisphere affected, and extent of injury), cognitive function, PWA assessor's profession, domains assessed by the PWA rating scale, and the quality of development and demonstration of the PWA rating scale, including sample size, sampling methods, sample representativeness, and blinding. Moreover, the table includes reported accuracy, sensitivity, specificity, reliability measures (e.g., internal consistency, retest reliability, and interrater reliability), and an overview of statistical procedures related to the PWA rating scale's validity (content validity, construct validity, and criterion-related validity). In the case of a diagnostic test to determine the presence/absence of aphasia, the table shall also present the scale used to validate the test and its cutoff values.

The table will include a summary and the reasons why the rating scale is appropriate for this study. We will adhere to the PRISMA-ScR guidelines [15].

### Stage 6: Consultation with reviewers and researchers

A stakeholder consultation will be held to examine the research findings. A speech-language pathologist with >10 years of experience in aphasia research will lead a team of two researchers. Their comments will be incorporated into how we present our final article.

## Discussion

Our scoping review will be significant in that it will collect data as thoroughly as possible, highlight data on the accuracy of rating scales of language function, communication ability, well-being, and QOL of PWA, and contribute to the enhancement of rehabilitation for PWA as well as the improvement of test accuracy. To summarize, this study will provide important data on which areas of the PWA rating scales are less accurate and the extent to which Japanese aphasic rating scales are less accurate. There have been no review studies of this type in both English-speaking countries and Japan, so we believe that this study is very novel.

## Supporting information

**S1 Checklist. Preferred Reporting Items for Systematic reviews and Meta-Analyses extension for Scoping Reviews (PRISMA-ScR) checklist.**
(DOCX)

**S1 File. Search strategy for PubMed electronic database.**
(DOCX)

## Author Contributions

**Investigation:** Satomi Kojima.

**Writing – original draft:** Yuhei Kodani.

**Writing – review & editing:** Shinsuke Nagami, Satomi Kojima, Shinya Fukunaga, Hikaru Nakamura.

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
