## [Decision Letter · Decision Letter 0]

8 Mar 2023

PONE-D-23-01103

Accuracy of rating scales for evaluating aphasic patient's psychological and language: A scoping review protocol

PLOS ONE

Dear Dr. Nagami,

Thank you for submitting your manuscript to PLOS ONE. After careful consideration, we feel that it has merit but does not fully meet PLOS ONE’s publication criteria as it currently stands. Therefore, we invite you to submit a revised version of the manuscript that addresses the points raised during the review process.

We look forward to receiving your revised manuscript.

Kind regards,

Gianmarco Abbadessa, MD

Academic Editor

PLOS ONE

Journal Requirements:

2. In your cover letter, please confirm that the research you have described in your manuscript, including participant recruitment, data collection, modification, or processing, has not started and will not start until after your paper has been accepted to the journal (assuming data need to be collected or participants recruited specifically for your study). In order to proceed with your submission, you must provide confirmation.

“We acknowledge that the power in the research from Grant-in-Aid for Research Activity Start-up (Grant Number 21K2115).”

“The funders had and will not have a role in study design, data collection and analysis, decision to publish, or preparation of the manuscript.”

“The funders had and will not have a role in study design, data collection and analysis, decision to publish, or preparation of the manuscript.”

a) If there are ethical or legal restrictions on sharing a de-identified data set, please explain them in detail (e.g., data contain potentially sensitive information, data are owned by a third-party organization, etc.) and who has imposed them (e.g., an ethics committee). Please also provide contact information for a data access committee, ethics committee, or other institutional body to which data requests may be sent. Please note that authors, including Corresponding Authors, are not permitted to be the sole point of contact for data requests.

b) If there are no restrictions, please provide the minimal anonymized data set necessary to replicate your study findings as either Supporting Information files or to a stable, public repository and provide us with the relevant URLs, DOIs, or accession numbers. For a list of acceptable repositories, please see http://journals.plos.org/plosone/s/data-availability#loc-recommended-repositories.

Reviewers' comments:

Reviewer's Responses to Questions

**Comments to the Author**

1. Does the manuscript provide a valid rationale for the proposed study, with clearly identified and justified research questions?

Reviewer #1: Yes

Reviewer #2: Yes

2. Is the protocol technically sound and planned in a manner that will lead to a meaningful outcome and allow testing the stated hypotheses?

Reviewer #1: Yes

Reviewer #2: Yes

3. Is the methodology feasible and described in sufficient detail to allow the work to be replicable?

Reviewer #1: Yes

Reviewer #2: Yes

4. Have the authors described where all data underlying the findings will be made available when the study is complete?

Reviewer #1: Yes

Reviewer #2: Yes

5. Is the manuscript presented in an intelligible fashion and written in standard English?

Reviewer #1: Yes

Reviewer #2: Yes

6. Review Comments to the Author

You may also provide optional suggestions and comments to authors that they might find helpful in planning their study.

Reviewer #1: The current paper entitled “Accuracy of rating scales for evaluating aphasic patient's psychological and language: A scoping review protocol ” proposes a scoping review protocol to evaluate the accuracy of the current rating scales for aphasia, by collecting articles (written in English and Japanese language) on this topic.

The paper is concise, well written, overall exhaustive and also the ultimate goal of the planned scoping review, i.e. to improve PWA rehabilitation, is undoubtedly relevant.

Nevertheless, I have some minor comments:

1)The authors often use accuracy, reliability and validity interchangeably (e.g. rows 89, 95, 113, 190), but this is sometimes misleading. Also, I suggest extracting and then reporting in a result table more measurement properties (other than reliability and validity) of the researched rating scales such as sensitivity, specificity and likelihood of a positive and a negative test. Additionally, it is appropriate to report details about these aspects (e.g. for reliability: both test-retest and inter-observer reliability; for validity its different forms such as construct, content and criterion as well as internal and external validity).

2) In the inclusion criteria I suggest adding a criterion for a minimum number of patients enrolled in each study (e.g. at least 20 patients for article).

3) In Table 1 under the column “exclusion criteria” primary progressive dementia is maybe primary progressive aphasia ?

4) In the section “Stage 4: Illustrating the results” I suggest to consider to extract and report other aspects that could influence the test validity and reliability such as the stroke type (ischemic vs hemorrhagic) and its characteristics (the affected hemisphere, the extent of the lesion, the existence of a previous important vascular disease that could reduce the functional reserve). In fact, reliability and validity may change on the basis of the number and types of the brain areas involved in a stroke. Also, it is extremely important taking into account the acute, subacute or chronic setting of the selected study, or, at least, reporting in table the elapsed time from the stroke and the test administration.

5) The authors will have to provide information on methodological quality for each validation study comprehensive of the reference test used, the specialist that assessed the aphasia-related features, the used cut-off and the presence or absence of blinding.

Some suggestions

--Although not mandatory for scoping reviews, the authors could consider to explain what type of quality assessment tool they intend to use (e.g. a checklist or a built-in function of Rayyan).

--Since low education levels may have an adverse effect on the result of language measurement, information about schooling/education (mean and relative range) of the studied populations could be useful.

--Since the author correctly sustain that assessment scales are usually chosen for subjective reasons, reporting the administration time of the used scale, the required specialization of the assessor (physicians, neurologist, speech experts, nurses etc.) and the availability of appropriate normative scores could provide relevant information about the feasibility, interpretability, acceptability of the rating scales.

Reviewer #2: The protocol is clear and well detailed: specifically the methods meet the state of the art standards. Therefore I do not suggest any revision

7. PLOS authors have the option to publish the peer review history of their article (what does this mean?). If published, this will include your full peer review and any attached files.

Reviewer #1: No

Reviewer #2: **Yes: **Andrea Plutino

---

## [Author Response · Author response to Decision Letter 0]

13 Apr 2023

PLOS ONE_ First revision_20230331

Upon scholarly discourse amongst the authors, the appellation of this manuscript underwent several refinements as articulated.

Accuracy of rating scales for evaluating aphasic patients’ psychological aspects and language function: A scoping review protocol

Reviewer 1 Comments

1) The authors often use accuracy, reliability and validity interchangeably (e.g. 

rows 89, 95, 113, 190), but this is sometimes misleading. Also, I suggest extracting and then reporting in a result table more measurement properties (other than reliability and validity) of the researched rating scales such as sensitivity, specificity and likelihood of a positive and a negative test. Additionally, it is appropriate to report details about these aspects (e.g. for reliability: both test-retest and inter-observer reliability; for validity its different forms such as construct, content and criterion as well as internal and external validity).

Thank you for your suggestions. We have revised the text as follows.

Line 78-81

The first purpose of this scoping review is to identify the domains of language function, communication ability, well-being, and QOL of PWA with low accuracy in rating scales based on data from the Japanese database (Ichushi web) and several international databases.

Line 83-86

Furthermore, the third purpose is to present differences in assessment scales for PWA in other countries and in Japan and to estimate the rehabilitation intervention for PWA in Japan and the accuracy of the assessment scales.

Line 195-198

Our scoping review will be significant in that it will collect data as thoroughly as possible, highlight data on the accuracy of rating scales of language function, communication ability, well-being, and QOL of PWA, and contribute to the enhancement of rehabilitation for PWA as well as the improvement of test accuracy.

Line 172-184

The presented table encompasses various essential aspects, such as the publication year, author’s name, pertinent PWA characteristics (e.g., age, gender, education level, length of time since stroke occurrence, stroke type, stroke frequency, hemisphere affected, and extent of injury), cognitive function, PWA assessor’s profession, domains assessed by the PWA rating scale, and the quality of development and demonstration of the PWA rating scale, including sample size, sampling methods, sample representativeness, and blinding. Moreover, the table includes reported accuracy, sensitivity, specificity, reliability measures (e.g., internal consistency, retest reliability, and interrater reliability), and an overview of statistical procedures related to the PWA rating scale’s validity (content validity, construct validity, and criterion-related validity). In the case of a diagnostic test to determine the presence/absence of aphasia, the table shall also present the scale used to validate the test and its cutoff values.

2) In the inclusion criteria I suggest adding a criterion for a minimum number of patients enrolled in each study (e.g. at least 20 patients for article).

Thank you for your suggestion. We have made the following corrections to Table 1 and additions to the text.

Table 1. Inclusion and exclusion criteria

Inclusion criteria Exclusion criteria

The literature is primarily concerned with rating scales aimed at not <20 individuals aged 18 years or older diagnosed with stroke-induced aphasia. This literature comprises observational and empirical studies, as well as research specifically aimed at developing, testing the reliability of and validating rating scales that assess various aspects of poststroke aphasic individuals' language functionality, communication abilities, overall well-being, and QOL. Literature focusing on aphasia from factors other than stroke (head trauma, Alzheimer's disease, cerebrovascular dementia, dementia with Lewy bodies, primary progressive aphasia, frontotemporal dementia, and progressive supranuclear palsy), literature also focusing on higher brain dysfunction other than aphasia, literature focusing on aphasic under 18 years old, literature focusing on aphasia syndromes of stroke, literature focusing on training for people with aphasia after stroke, literature published before 1995, literature in languages other than English and Japanese, intervention studies, scoping reviews, systemic reviews, meta-analysis, case studies, and letter papers.

Line 135-138

The first inclusion criterion is the exclusive incorporation of rating scales duly validated with a minimum sample size of 20 participants with PWA aged 18 years or above and supplementation by a cohort with aphasia.

3) In Table 1 under the column “exclusion criteria” primary progressive dementia is maybe primary progressive aphasia ?

Thank you for your careful review. We have revised Table 1 as follows

Table 1. Inclusion and exclusion criteria

Inclusion criteria 

The literature is primarily concerned with rating scales aimed at not <20 individuals aged 18 years or older diagnosed with stroke-induced aphasia. This literature comprises observational and empirical studies, as well as research specifically aimed at developing, testing the reliability of and validating rating scales that assess various aspects of poststroke aphasic individuals' language functionality, communication abilities, overall well-being, and QOL.

Exclusion criteria

Literature focusing on aphasia from factors other than stroke (head trauma, Alzheimer's disease, cerebrovascular dementia, dementia with Lewy bodies, primary progressive aphasia, frontotemporal dementia, and progressive supranuclear palsy), literature also focusing on higher brain dysfunction other than aphasia, literature focusing on aphasic under 18 years old, literature focusing on aphasia syndromes of stroke, literature focusing on training for people with aphasia after stroke, literature published before 1995, literature in languages other than English and Japanese, intervention studies, scoping reviews, systemic reviews, meta-analysis, case studies, and letter papers.

4) In the section “Stage 4: Illustrating the results” I suggest to consider to extract and report other aspects that could influence the test validity and reliability such as the stroke type (ischemic vs hemorrhagic) and its characteristics (the affected hemisphere, the extent of the lesion, the existence of a previous important vascular disease that could reduce the functional reserve). In fact, reliability and validity may change on the basis of the number and types of the brain areas involved in a stroke. Also, it is extremely important taking into account the acute, subacute or chronic setting of the selected study, or, at least, reporting in table the elapsed time from the stroke and the test administration.

5) The authors will have to provide information on methodological quality for each validation study comprehensive of the reference test used, the specialist that assessed the aphasia-related features, the used cut-off and the presence or absence of blinding.

Thank you for your suggestion. We have added the following text in the revised manuscript.

Line 156-164

The obtained results encompass a wide array of information, including the author's name, publication year, study title, country of origin, study purpose, sampling method, sample characteristics (sex, age, educational background, presence/absence of prior stroke, and cognitive assessment results), stroke characteristics (stroke type, hemisphere affected, and extent and duration of injury), sample size, PWA assessor’s profession, rating scale utilized for PWA evaluation, rating scale domains, details of the rating scale development process, information on rating scale sensitivity and specificity, and information related to the reviews of the rating scale’s reliability and validity.

Line 172-184

The presented table encompasses various essential aspects, such as the publication year, author’s name, pertinent PWA characteristics (e.g., age, gender, education level, length of time since stroke occurrence, stroke type, stroke frequency, hemisphere affected, and extent of injury), cognitive function, PWA assessor’s profession, domains assessed by the PWA rating scale, and the quality of development and demonstration of the PWA rating scale, including sample size, sampling methods, sample representativeness, and blinding. Moreover, the table includes reported accuracy, sensitivity, specificity, reliability measures (e.g., internal consistency, retest reliability, and interrater reliability), and an overview of statistical procedures related to the PWA rating scale’s validity (content validity, construct validity, and criterion-related validity). In the case of a diagnostic test to determine the presence/absence of aphasia, the table shall also present the scale used to validate the test and its cutoff values.

---

## [Editor Report · Decision Letter 1]

28 Apr 2023

Accuracy of rating scales for evaluating aphasic patients’ psychological aspects and language function: A scoping review protocol

PONE-D-23-01103R1

Dear Dr. Nagami,

I'm pleased to inform you that, based on the responses provided to the reviewers concerns that, in my opinion, results addressed, your manuscript can be considered scientifically suitable for publication. It will be formally accepted for publication once it meets all outstanding technical requirements..

Kind regards,

Gianmarco Abbadessa, MD

Academic Editor

PLOS ONE
---

## [Editor Report · Acceptance letter]

7 May 2023

PONE-D-23-01103R1 

Accuracy of rating scales for evaluating aphasic patients’ psychological aspects and language function: A scoping review protocol 

Dear Dr. Nagami:

I'm pleased to inform you that your manuscript has been deemed suitable for publication in PLOS ONE. Congratulations! Your manuscript is now with our production department. 

Kind regards, 

on behalf of

Dr. Gianmarco Abbadessa 

Academic Editor

PLOS ONE